# Self-bound droplets in quasi-two-dimensional dipolar condensates

Yuqi Wang,[1] Tao Shi,[1, 2, 3, *] and Su Yi[1, 2, 3, †]

[1]*CAS Key Laboratory of Theoretical Physics, Institute of Theoretical Physics,*
*Chinese Academy of Sciences, Beijing 100190, China*
[2]*CAS Center for Excellence in Topological Quantum Computation & School of Physical Sciences,*
*University of Chinese Academy of Sciences, Beijing 100049, China*
[3]*Peng Huanwu Collaborative Center for Research and Education, Beihang University, Beijing 100191, China*
(Dated: October 28, 2022)

We study the ground-state properties of self-bound dipolar droplets in quasi-two-dimensional geometry by using the Gaussian state theory. We show that there exist two quantum phases corresponding to the macroscopic squeezed vacuum and squeezed coherent states. We further show that the radial size versus atom number curve exhibits a double-dip structure, as a result of the multiple quantum phases. In particular, we find that the critical atom number for the self-bound droplets is determined by the quantum phases, which allows us to distinguish the quantum state and validates the Gaussian state theory.

## I. INTRODUCTION

Quantum droplets of atomic Bose-Einstein condensates have attracted tremendous interests recently [1–7]. Extensive experiments have been devoted to studying their fascinating properties such as soliton to droplet crossover [8], supersolid states [9–11], collective excitation [12, 13], Goldstone mode [14], and collision dynamics [15]. Theoretically, because droplets are found in the mean-field collapse regime, understanding their stabilization mechanism is of fundamental importance. Currently, it is widely accepted that self-bound droplets are stablized by quantum fluctuations [16–19]. In the weak interaction regime, quantum fluctuation can be incorporated into the Gross-Pitaevskii equation through the Lee-Huang-Yang correction [20] and leads to the extended Gross-Pitaevskii equation (EGPE) [19, 21, 22]. Although the EGPE provided satisfactory explanations to some experiments, there are still discrepancies between the predictions from EGPE and experiments [4, 23].

Theories beyond EGPE, such as quantum Monte Carlo method [23–28] and time-dependent Hatree-Fock-Bogoliubov equations [29], are employed to study the properties of the self-bound droplets. Higher order corrections to the EGPE have also been developed [30–32]. In particular, we systematically studied the self-bound droplets of dipolar and binary condensates [33, 34] using the Gaussian state theory (GST) [35, 36], a generalization to the conventional coherent-state-based mean-field theory. In GST, quantum fluctuation is inherently included in the many-body wave function and is treated self-consistently. Compared to the well-known Hartree-Fock-Bogoliubov theory, this feature makes it particularly convenient for characterizing the quantum states of a condensate. In addition, GST circumvents the difficulty in the Hartree-Fock-Bogoliubov theory (HFBT) by pro-

ducing the gapless excitation spectrum of condensates, as required by Goldstone theorem for the U(1) symmetry breaking [36–38].

When applied to study the ground-state properties of a trapped condensate with $s$-wave attraction, the GST showed that the ground state of the system is a macroscopic single-mode squeezed vacuum state (SVS) [37], in contrast to the commonly believed coherent state (CS). Physically, this can be understood by noting that the interaction energy for SVS contains contributions from the Hartree, Fock, and pairing terms, as compared to the single Hartree term for CS. Energetically, the SVS is more favorable when the $s$-wave scattering length is negative.

In the presence of the three-body repulsion, we also studied the quantum phases of the self-bound droplets of both dipolar [33] and quasi-2D binary condensates [34]. We showed that, due to the competition between the 2B attraction and the 3B repulsion, the squeezed coherent state (SCS) and the coherent state (CS) phases also emerge as condensate density increases. Of particular interest, in quasi-2D binary droplets, we found various signatures on the radial size ($\sigma$) versus atom number ($N$) curve that can be used to identify the quantum states and the stabilization mechanisms. Furthermore, because the 2B attractive interaction can be canceled out exactly by the kinetic energy in 2D geometry, the critical atom number (CAN), the minimal number of atoms needed to form a self-bound state, is independent of the stabilization mechanism. Instead, precise measurement of the CAN allows us to distinguish the quantum state of the droplets.

An immediate question to ask is whether it is possible to generalize our studies to dipolar condensates. In this work, we propose to generate the self-bound quasi-2D dipolar droplets by tuning dipolar interaction. We then study the ground-state properties of system via the GST. We show that, in quasi-2D geometry, the self-bound state only contains the SVS and the SCS phases. The CS phase which exists in the 3D case is now missing even in the strong interaction regime. Interestingly, similar to that in the binary droplets, we also find the W-shaped $\sigma$-$N$

* tshi@itp.ac.cn
† syi@itp.ac.cn

curve which manifests the existence of multiple quantum phase in the self-bound droplet state. In addition, the CAN is solely determined by the quantum states of the condensate. As a result, the precise measurement of the CAN allows us to distinguish squeezed state from coherent one.

The rest of this paper is organized as follows. In Sec. II, we introduce the quasi-2D model and the Gaussian state theory. In Sec. III, after validating our quasi-2D treatment, we present our results on the ground-state properties of the quasi-2D dipolar droplets, including quantum phases, peak density, and radial size. Finally, we conclude in Sec. IV.

## II. FORMULATION

### A. Model

We consider an ultracold gas of $N$ interacting $^{162}$Dy atoms confined in a one-dimensional harmonic trap. In second-quantized form, the Hamiltonian of the system reads

$$
\hat{H}^{(3D)} = \int d\mathbf{r}\hat{\psi}^\dagger(\mathbf{r}) \left[ -\frac{\hbar^2\nabla^2}{2M} + V(\mathbf{r}) - \mu \right] \hat{\psi}(\mathbf{r})
$$
$$
+ \frac{1}{2} \int d\mathbf{r}d\mathbf{r}'\hat{\psi}^\dagger(\mathbf{r})\hat{\psi}^\dagger(\mathbf{r}')U^{(3D)}(\mathbf{r}-\mathbf{r}')\hat{\psi}(\mathbf{r}')\hat{\psi}(\mathbf{r})
$$
$$
+ \frac{g_3}{3!} \int d\mathbf{r}\hat{\psi}^{\dagger 3}(\mathbf{r})\hat{\psi}^3(\mathbf{r}), \qquad (1)
$$

where $\hat{\psi}^\dagger(\mathbf{r})$ is the field operator, $M$ is the mass of the atom, $V(\mathbf{r}) = M\omega_z^2 z^2/2$ is the 1D harmonic trap with $\omega_z$ being the trap frequency, and $\mu$ is the chemical potential. The two-body interaction potential consists of the contact and the dipole-dipole interaction (DDI), i.e.,

$$
U^{(3D)}(\mathbf{r}) = \frac{4\pi\hbar^2 a_s}{M}\delta(\mathbf{r}) + \frac{\mu_0 d^2}{4\pi}\frac{1-3\cos^2\theta_\mathbf{r}}{|\mathbf{r}|^3}, \quad (2)
$$

where $a_s$ is the $s$-wave scattering length, $\mu_0$ is the vacuum permeability, $d = 9.93\,\mu_B$ is the magnetic dipole moment of Dy atom with $\mu_B$ the Bohr magneton, and $\theta_\mathbf{r}$ is the polar angle of $\mathbf{r}$. It should be noted that, without loss of generality, we have assumed that the dipole moment is polarized along the $z$ axis in Eq. (2). The last

term of Eq. (1) represents the three-body (3B) interaction with $g_3$ being the 3B coupling strength. To stabilize the condensate, we always assume that $g_3 > 0$.

In this work, we focus on the quasi-2D geometry which can be realized when $\omega_z$ is sufficiently large such that the motion of atoms along the $z$ axis is frozen to the ground state of the harmonic oscillator. As a result, the field operator can be factorized into

$$
\hat{\psi}(\mathbf{r}) = \hat{\psi}(\boldsymbol{\rho})\frac{e^{-z^2/(2a_z^2)}}{(\pi a_z^2)^{1/4}}, \qquad (3)
$$

where $\boldsymbol{\rho} \equiv (x,y)$ and $a_z = \sqrt{\hbar/(M\omega_z)}$ is the harmonic oscillator length. After integrating out the $z$ variable, Hamiltonian (1) reduces to

$$
\hat{H} = \int d\boldsymbol{\rho}\hat{\psi}^\dagger(\boldsymbol{\rho})h_0(\boldsymbol{\rho})\hat{\psi}(\boldsymbol{\rho})
$$
$$
+ \frac{1}{2} \int d\boldsymbol{\rho}d\boldsymbol{\rho}'\hat{\psi}^\dagger(\boldsymbol{\rho})\hat{\psi}^\dagger(\boldsymbol{\rho}')U(\boldsymbol{\rho}-\boldsymbol{\rho}')\hat{\psi}(\boldsymbol{\rho}')\hat{\psi}(\boldsymbol{\rho})
$$
$$
+ \frac{g_3}{3!\sqrt{3}\pi a_z^2} \int d\boldsymbol{\rho}\hat{\psi}^{\dagger 3}(\boldsymbol{\rho})\hat{\psi}^3(\boldsymbol{\rho}), \qquad (4)
$$

where $h_0 = -\hbar^2(\partial_x^2+\partial_y^2)/(2M)-\mu$ with an unimportant constant being absorbed into the chemical potential. The effective 2D interaction potential is

$$
U(\boldsymbol{\rho}) = \frac{4\pi\hbar^2 a_s}{\sqrt{2\pi}a_z M}\delta(\boldsymbol{\rho}) + \mathcal{F}^{-1}[\widetilde{U}_{dd}], \qquad (5)
$$

where $\mathcal{F}^{-1}[\cdot]$ denotes the inverse Fourier transform and

$$
\widetilde{U}_{dd}(\boldsymbol{k}_\rho) = \frac{4\pi\hbar^2 a_{dd}}{\sqrt{2\pi}a_z M}D\left(\frac{|\boldsymbol{k}_\rho|a_z}{\sqrt{2}}\right) \qquad (6)
$$

is the effective 2D dipolar interaction in the 2D momentum space with $\mathbf{k}_\rho \equiv (k_x, k_y)$ [39]. Here $D(x) = 2-3\sqrt{\pi}xe^{x^2}\text{erfc}(x)$ with $\text{erfc}(x)$ being the complementary error function and $a_{dd} = \mu_0 d^2 M/(12\pi\hbar^2)$ is the dipolar interaction length. For $^{162}$Dy atom, the dipolar interaction length is $a_{dd} = 129\,a_B$ with $a_B$ being the Bohr radius.

### B. Gaussian state theory

To be self-contained, let us briefly introduce GST [33–35, 37]. Going beyond the coherent-state-based Gross-Pitaevskii equation description of the Bose-Einstein condensates, we approximate the many-body ground state of our systems by a pure Gaussian state (or coherent-squeezed state), i.e.,

$$
|\text{GS}\rangle = \exp\left\{\int d\boldsymbol{\rho}\left[\hat{\psi}^\dagger(\boldsymbol{\rho})\phi(\boldsymbol{\rho}) - \hat{\psi}(\boldsymbol{\rho})\phi^*(\boldsymbol{\rho})\right]\right\}\exp\left\{\frac{i}{2}\int d\boldsymbol{\rho}d\boldsymbol{\rho}'\left[\hat{\psi}^\dagger(\boldsymbol{\rho})\xi(\boldsymbol{\rho},\boldsymbol{\rho}')\hat{\psi}^\dagger(\boldsymbol{\rho}') + \hat{\psi}(\boldsymbol{\rho})\xi^*(\boldsymbol{\rho},\boldsymbol{\rho}')\hat{\psi}(\boldsymbol{\rho}')\right]\right\}|0\rangle, \quad (7)
$$

where $\phi(\boldsymbol{\rho})$ and $\xi(\boldsymbol{\rho}, \boldsymbol{\rho}')$ are the variational parameters to be determined. It can be easily shown that $\phi(\boldsymbol{\rho}) = \langle \mathrm{GS}|\hat{\psi}(\boldsymbol{\rho})|\mathrm{GS}\rangle$ is the mean value of the field operator and represents the coherent part of the state. Furthermore, $\xi(\boldsymbol{\rho}, \boldsymbol{\rho}')$ is symmetric, i.e., $\xi(\boldsymbol{\rho}', \boldsymbol{\rho}) = \xi(\boldsymbol{\rho}, \boldsymbol{\rho}')$, and characterizes the squeezed part of the state.

From a physical point of view, we usually use the normal and anomalous correlation functions, i.e., $G(\boldsymbol{\rho}, \boldsymbol{\rho}') = \langle \delta\hat{\psi}^{\dagger}(\boldsymbol{\rho}')\delta\hat{\psi}(\boldsymbol{\rho})\rangle$ and $F(\boldsymbol{\rho}, \boldsymbol{\rho}') = \langle \delta\hat{\psi}(\boldsymbol{\rho})\delta\hat{\psi}(\boldsymbol{\rho}')\rangle$, respectively, to describe the squeezing of the system, where $\delta\hat{\psi}(\boldsymbol{\rho}) = \hat{\psi}(\boldsymbol{\rho}) - \phi(\boldsymbol{\rho})$ is the fluctuation operator. The correlation functions $G, F$ and the squeezed parameter $\xi$ are connected through the covariance matrix $\Gamma(\boldsymbol{\rho}, \boldsymbol{\rho}') \equiv \langle \{\delta\hat{\Psi}(\boldsymbol{\rho}), \delta\hat{\Psi}^{\dagger}(\boldsymbol{\rho}')\}\rangle$ by the relation

$$\begin{pmatrix} 2G(\boldsymbol{\rho}, \boldsymbol{\rho}') + \delta(\boldsymbol{\rho} - \boldsymbol{\rho}') & 2F(\boldsymbol{\rho}, \boldsymbol{\rho}') \\ 2F^*(\boldsymbol{\rho}, \boldsymbol{\rho}') & 2G^*(\boldsymbol{\rho}, \boldsymbol{\rho}') + \delta(\boldsymbol{\rho} - \boldsymbol{\rho}') \end{pmatrix} = \Gamma(\boldsymbol{\rho}, \boldsymbol{\rho}') = \int d\boldsymbol{\rho}'' S(\boldsymbol{\rho}, \boldsymbol{\rho}'') S^{\dagger}(\boldsymbol{\rho}'', \boldsymbol{\rho}'), \tag{8}$$

where $\delta\hat{\Psi}(\boldsymbol{\rho}) = \begin{pmatrix} \delta\hat{\psi}(\boldsymbol{\rho}) \\ \delta\hat{\psi}^{\dagger}(\boldsymbol{\rho}) \end{pmatrix}$ is the fluctuation operator expressed in the Nambu basis, and

$$S(\boldsymbol{\rho}, \boldsymbol{\rho}') = \exp\left\{ \begin{pmatrix} 0 & i\xi(\boldsymbol{\rho}, \boldsymbol{\rho}') \\ -i\xi^*(\boldsymbol{\rho}, \boldsymbol{\rho}') & 0 \end{pmatrix} \right\}. \tag{9}$$

Now, finding the ground-state wave function then reduces to obtaining the variational parameters $\phi$, $G$, and $F$ that minimizes the total energy of the system.

In GST, the ground-state variational parameters are obtained by evolving the imaginary-time equations of motion (ITEOM) for $\phi, G$ and $F$. To derive these equations, we substitute $\hat{\psi} = \phi + \delta\hat{\psi}$ into the Hamiltonian (4) and apply the Wick's theorem such that, up to the quadratic order in $\delta\hat{\psi}$, the Hamiltonian $\hat{H}$ reduces to the mean-field Hamiltonian

$$\hat{H}_{\mathrm{MF}} = E + \int d\boldsymbol{\rho}\left[\delta\hat{\psi}^{\dagger}(\boldsymbol{\rho})\eta(\boldsymbol{\rho}) + \mathrm{h.c.}\right] + \frac{1}{2}\int d\boldsymbol{\rho} d\boldsymbol{\rho}' :\delta\hat{\Psi}^{\dagger}(\boldsymbol{\rho})\mathcal{H}(\boldsymbol{\rho}, \boldsymbol{\rho}')\delta\hat{\Psi}(\boldsymbol{\rho}'):, \tag{10}$$

where $:\hat{O}:$ denotes the normal ordered operator with respect to the Gaussian state $|\mathrm{GS}\rangle$,

$$\eta(\boldsymbol{\rho}) = h_0(\boldsymbol{\rho})\phi(\boldsymbol{\rho}) + \int d\boldsymbol{\rho}' U(\boldsymbol{\rho} - \boldsymbol{\rho}')\left\{\phi(\boldsymbol{\rho})\left[|\phi(\boldsymbol{\rho}')|^2 + G(\boldsymbol{\rho}', \boldsymbol{\rho}')\right] + \phi(\boldsymbol{\rho}')G(\boldsymbol{\rho}, \boldsymbol{\rho}') + \phi^*(\boldsymbol{\rho}')F(\boldsymbol{\rho}, \boldsymbol{\rho}')\right\}$$

$$+ \frac{g_3}{2\sqrt{3}\pi a_z^2}\left\{\left[|\phi(\boldsymbol{\rho})|^4 + \phi^2(\boldsymbol{\rho})F^*(\boldsymbol{\rho}, \boldsymbol{\rho}) + 6|\phi(\boldsymbol{\rho})|^2 G(\boldsymbol{\rho}, \boldsymbol{\rho}) + 3|F(\boldsymbol{\rho}, \boldsymbol{\rho})|^2 + 6G^2(\boldsymbol{\rho}, \boldsymbol{\rho})\right]\phi(\boldsymbol{\rho})\right.$$

$$\left. + 3\left[|\phi(\boldsymbol{\rho})|^2 + 2G(\boldsymbol{\rho}, \boldsymbol{\rho})\right]F(\boldsymbol{\rho}, \boldsymbol{\rho})\phi^*(\boldsymbol{\rho})\right\} \tag{11}$$

is the linear driving term with $G(\boldsymbol{\rho}, \boldsymbol{\rho}') = \langle \delta\hat{\psi}^{\dagger}(\boldsymbol{\rho}')\delta\hat{\psi}(\boldsymbol{\rho})\rangle$ and $F(\boldsymbol{\rho}, \boldsymbol{\rho}') = \langle \delta\hat{\psi}(\boldsymbol{\rho})\delta\hat{\psi}(\boldsymbol{\rho}')\rangle$ being the normal and the anomalous Green functions, respectively, and $\mathcal{H}(\boldsymbol{\rho}, \boldsymbol{\rho}') = \begin{pmatrix} \mathcal{E}(\boldsymbol{\rho}, \boldsymbol{\rho}') & \Delta(\boldsymbol{\rho}, \boldsymbol{\rho}') \\ \Delta^*(\boldsymbol{\rho}, \boldsymbol{\rho}') & \mathcal{E}^*(\boldsymbol{\rho}, \boldsymbol{\rho}') \end{pmatrix}$ is the mean-field Hamiltonian with

$$\mathcal{E}(\boldsymbol{\rho}, \boldsymbol{\rho}') = h_0(\boldsymbol{\rho})\delta(\boldsymbol{\rho} - \boldsymbol{\rho}') + \delta(\boldsymbol{\rho} - \boldsymbol{\rho}')\int d\boldsymbol{\rho}'' U(\boldsymbol{\rho} - \boldsymbol{\rho}'')\left[|\phi(\boldsymbol{\rho}'')|^2 + G(\boldsymbol{\rho}'', \boldsymbol{\rho}'')\right] + U(\boldsymbol{\rho} - \boldsymbol{\rho}')\left[\phi^*(\boldsymbol{\rho}')\phi(\boldsymbol{\rho}) + G(\boldsymbol{\rho}, \boldsymbol{\rho}')\right]$$

$$+ \frac{3g_3}{2\sqrt{3}\pi a_z^2}\delta(\boldsymbol{\rho} - \boldsymbol{\rho}')\left[|\phi(\boldsymbol{\rho})|^4 + 4|\phi(\boldsymbol{\rho})|^2 G(\boldsymbol{\rho}, \boldsymbol{\rho}) + 2G^2(\boldsymbol{\rho}, \boldsymbol{\rho}) + \phi^{*2}(\boldsymbol{\rho})F(\boldsymbol{\rho}, \boldsymbol{\rho}) + \phi^2(\boldsymbol{\rho})F^*(\boldsymbol{\rho}, \boldsymbol{\rho}) + |F(\boldsymbol{\rho}, \boldsymbol{\rho})|^2\right], \tag{12a}$$

$$\Delta(\boldsymbol{\rho}, \boldsymbol{\rho}') = U(\boldsymbol{\rho} - \boldsymbol{\rho}')\left[\phi(\boldsymbol{\rho})\phi(\boldsymbol{\rho}') + F(\boldsymbol{\rho}, \boldsymbol{\rho}')\right]$$

$$+ \frac{g_3}{\sqrt{3}\pi a_z^2}\delta(\boldsymbol{\rho} - \boldsymbol{\rho}')\left[|\phi(\boldsymbol{\rho})|^2\phi^2(\boldsymbol{\rho}) + 3\phi^2(\boldsymbol{\rho})G(\boldsymbol{\rho}, \boldsymbol{\rho}) + 3|\phi(\boldsymbol{\rho})|^2 F(\boldsymbol{\rho}, \boldsymbol{\rho}) + 3G(\boldsymbol{\rho}, \boldsymbol{\rho})F(\boldsymbol{\rho}, \boldsymbol{\rho})\right]. \tag{12b}$$

Moreover, $E = \langle \mathrm{GS}|\hat{H}|\mathrm{GS}\rangle = E_{\mathrm{kin}} + E_{2B} + E_{3B}$ in Eq. (10) is the total energy which consists of the kinetic energy

($E_{\text{kin}}$), the 2B interaction energy ($E_{2B}$), and the 3B interaction energy ($E_{3B}$). In explicit form, we have

$$E_{\text{kin}} = \int d\boldsymbol{\rho}\big[\phi^*(\boldsymbol{\rho})h_0(\boldsymbol{\rho})\phi(\boldsymbol{\rho}) + \lim_{\boldsymbol{\rho}'\to\boldsymbol{\rho}} h_0(\boldsymbol{\rho})G(\boldsymbol{\rho},\boldsymbol{\rho}')\big], \tag{13a}$$

$$E_{2B} = \int d\boldsymbol{\rho}d\boldsymbol{\rho}'U(\boldsymbol{\rho}-\boldsymbol{\rho}')\Big\{\frac{1}{2}|\phi(\boldsymbol{\rho})|^2|\phi(\boldsymbol{\rho}')|^2 + G(\boldsymbol{\rho},\boldsymbol{\rho})\big[|\phi(\boldsymbol{\rho}')|^2 + \frac{1}{2}G(\boldsymbol{\rho}',\boldsymbol{\rho}')\big]$$
$$+ G(\boldsymbol{\rho}',\boldsymbol{\rho})\big[\phi(\boldsymbol{\rho})\phi^*(\boldsymbol{\rho}') + \frac{1}{2}G(\boldsymbol{\rho},\boldsymbol{\rho}')\big] + \text{Re}F^*(\boldsymbol{\rho},\boldsymbol{\rho}')\big[\phi(\boldsymbol{\rho})\phi(\boldsymbol{\rho}') + \frac{1}{2}F(\boldsymbol{\rho},\boldsymbol{\rho}')\big]\Big\}, \tag{13b}$$

$$E_{3B} = \frac{g_3}{\sqrt{3}\pi a_z^2}\int d\boldsymbol{\rho}\Big[\frac{1}{3!}|\phi(\boldsymbol{\rho})|^6 + \frac{3}{2}|\phi(\boldsymbol{\rho})|^4 G(\boldsymbol{\rho},\boldsymbol{\rho}) + \text{Re}|\phi(\boldsymbol{\rho})|^2\phi^2(\boldsymbol{\rho})F^*(\boldsymbol{\rho},\boldsymbol{\rho}) + 3\text{Re}\phi^2(\boldsymbol{\rho})F^*(\boldsymbol{\rho},\boldsymbol{\rho})G(\boldsymbol{\rho},\boldsymbol{\rho})$$
$$+ 3|\phi(\boldsymbol{\rho})|^2 G^2(\boldsymbol{\rho},\boldsymbol{\rho}) + \frac{3}{2}|\phi(\boldsymbol{\rho})|^2|F(\boldsymbol{\rho},\boldsymbol{\rho})|^2 + \frac{3}{2}G(\boldsymbol{\rho},\boldsymbol{\rho})|F(\boldsymbol{\rho},\boldsymbol{\rho})|^2 + G^3(\boldsymbol{\rho},\boldsymbol{\rho})\Big]. \tag{13c}$$

The ITEOM can be obtained by minimizing the total energy with respect to $\phi$, $G$, and $F$. After projecting the resulting equations onto the tangential space of the Gaussian manifold [33–35, 37], we obtain

$$\partial_\tau\phi(\boldsymbol{\rho}) = -\eta(\boldsymbol{\rho}) - 2\int d\boldsymbol{\rho}'\Big[G(\boldsymbol{\rho},\boldsymbol{\rho}')\eta(\boldsymbol{\rho}') + F^*(\boldsymbol{\rho},\boldsymbol{\rho}')\eta^*(\boldsymbol{\rho}')\Big], \tag{14a}$$

$$\partial_\tau G(\boldsymbol{\rho},\boldsymbol{\rho}') = -\int d\boldsymbol{\rho}''\Big[G(\boldsymbol{\rho},\boldsymbol{\rho}'')\mathcal{E}(\boldsymbol{\rho}'',\boldsymbol{\rho}') + F(\boldsymbol{\rho},\boldsymbol{\rho}'')\Delta^*(\boldsymbol{\rho}'',\boldsymbol{\rho}') + \mathcal{E}(\boldsymbol{\rho},\boldsymbol{\rho}'')G(\boldsymbol{\rho}'',\boldsymbol{\rho}') + \Delta(\boldsymbol{\rho},\boldsymbol{\rho}'')F^*(\boldsymbol{\rho}'',\boldsymbol{\rho}')\Big]$$
$$- 2\int d\boldsymbol{\rho}''d\boldsymbol{\rho}'''\Big[G(\boldsymbol{\rho},\boldsymbol{\rho}'')\mathcal{E}(\boldsymbol{\rho}'',\boldsymbol{\rho}''')G(\boldsymbol{\rho}''',\boldsymbol{\rho}') + F(\boldsymbol{\rho},\boldsymbol{\rho}'')\Delta^*(\boldsymbol{\rho}'',\boldsymbol{\rho}''')G(\boldsymbol{\rho}''',\boldsymbol{\rho}')$$
$$+ G(\boldsymbol{\rho},\boldsymbol{\rho}'')\Delta(\boldsymbol{\rho}'',\boldsymbol{\rho}''')F^*(\boldsymbol{\rho}''',\boldsymbol{\rho}') + F(\boldsymbol{\rho},\boldsymbol{\rho}'')\mathcal{E}^*(\boldsymbol{\rho}'',\boldsymbol{\rho}''')F^*(\boldsymbol{\rho}''',\boldsymbol{\rho}')\Big], \tag{14b}$$

$$\partial_\tau F(\boldsymbol{\rho},\boldsymbol{\rho}') = -\Delta(\boldsymbol{\rho},\boldsymbol{\rho}') - \int d\boldsymbol{\rho}''\Big[G(\boldsymbol{\rho},\boldsymbol{\rho}'')\Delta(\boldsymbol{\rho}'',\boldsymbol{\rho}') + F(\boldsymbol{\rho},\boldsymbol{\rho}'')\mathcal{E}^*(\boldsymbol{\rho}'',\boldsymbol{\rho}') + \mathcal{E}(\boldsymbol{\rho},\boldsymbol{\rho}'')F(\boldsymbol{\rho}'',\boldsymbol{\rho}') + \Delta(\boldsymbol{\rho},\boldsymbol{\rho}'')G^*(\boldsymbol{\rho}'',\boldsymbol{\rho}')\Big]$$
$$- 2\int d\boldsymbol{\rho}''d\boldsymbol{\rho}'''\Big[G(\boldsymbol{\rho},\boldsymbol{\rho}'')\mathcal{E}(\boldsymbol{\rho}'',\boldsymbol{\rho}''')F(\boldsymbol{\rho}''',\boldsymbol{\rho}') + F(\boldsymbol{\rho},\boldsymbol{\rho}'')\Delta^*(\boldsymbol{\rho}'',\boldsymbol{\rho}''')F(\boldsymbol{\rho}''',\boldsymbol{\rho}')$$
$$+ G(\boldsymbol{\rho},\boldsymbol{\rho}'')\Delta(\boldsymbol{\rho}'',\boldsymbol{\rho}''')G^*(\boldsymbol{\rho}''',\boldsymbol{\rho}') + F(\boldsymbol{\rho},\boldsymbol{\rho}'')\mathcal{E}^*(\boldsymbol{\rho}'',\boldsymbol{\rho}''')G^*(\boldsymbol{\rho}''',\boldsymbol{\rho}')\Big]. \tag{14c}$$

Numerically, one may evolve Eqs. (14) simultaneously, which eventually converges to the order parameters for the ground state.

The physical significance of the Gaussian-state wave function can be revealed by factorizing $|\text{GS}\rangle$. To this end, we first note that, as a symmetric matrix, $\xi(\boldsymbol{\rho},\boldsymbol{\rho}')$ can be Autonne-Takagi diagonalized by a set of orthonormal functions $\{\bar{\phi}_{s,j}\}$ as [40, 41]

$$\xi(\boldsymbol{\rho},\boldsymbol{\rho}') = \sum_{j=1}^\infty d_j\bar{\phi}_{s,j}(\boldsymbol{\rho})\bar{\phi}_{s,j}(\boldsymbol{\rho}'), \tag{15}$$

where $\int d\boldsymbol{\rho}\bar{\phi}_{s,i}^*(\boldsymbol{\rho})\bar{\phi}_{s,j}(\boldsymbol{\rho}) = \delta_{ij}$ and $d_j$ are real. As a result, the Gaussian state can be reexpressed in the form of the familiar multi-mode squeezed-coherent state [33, 34], i.e.,

$$|\text{GS}\rangle = e^{\sqrt{N_c}(\hat{a}^\dagger-\hat{a})}\prod_{j=1}^\infty e^{\frac{1}{2}d_j(\hat{b}_j^{\dagger 2}-\hat{b}_j^2)}|0\rangle, \tag{16}$$

where $\hat{a} = \int d\boldsymbol{\rho}\bar{\phi}_c^*(\boldsymbol{\rho})\hat{\psi}(\boldsymbol{\rho})$ is the annihilation operator for the coherent mode with $\bar{\phi}_c(\boldsymbol{\rho}) = \phi(\boldsymbol{\rho})/\sqrt{N_c}$ being the normalized coherent mode function and $N_c =$

$\int d\boldsymbol{\rho}|\phi(\boldsymbol{\rho})|^2$ the occupation number in the coherent mode. Moreover, $\hat{b}_j = \int d\boldsymbol{\rho}\bar{\phi}_{s,j}^*(\boldsymbol{\rho})\hat{\psi}(\boldsymbol{\rho})$ is the annihilation operator of the $j$th squeezed mode with $N_{s,j} = \sinh^2 d_j$ being the corresponding occupation number. Without loss of generality, we assume that $N_{s,j}$ are sorted in the descending order with respect to the index $j$. The total number of the squeezed atoms is then $N_s = \sum_j N_{s,j}$. The correlation functions $G$ and $F$ can also be diagonalized as [33, 34]

$$G(\boldsymbol{\rho},\boldsymbol{\rho}') = \sum_{j=1}^\infty N_{s,j}\bar{\phi}_{s,j}(\boldsymbol{\rho})\bar{\phi}_{s,j}^*(\boldsymbol{\rho}'), \tag{17a}$$

$$F(\boldsymbol{\rho},\boldsymbol{\rho}') = \sum_{j=1}^\infty \sqrt{N_{s,j}(N_{s,j}+1)}\bar{\phi}_{s,j}(\boldsymbol{\rho})\bar{\phi}_{s,j}(\boldsymbol{\rho}'). \tag{17b}$$

The total number of atoms

$$N = \int d\boldsymbol{\rho}\langle\hat{\psi}^\dagger(\boldsymbol{\rho})\hat{\psi}(\boldsymbol{\rho})\rangle = \int d\boldsymbol{\rho}\Big[|\phi(\boldsymbol{\rho})|^2 + G(\boldsymbol{\rho},\boldsymbol{\rho})\Big]$$
$$= N_c + N_s \tag{18}$$

is composed of the coherent and squeezed parts.

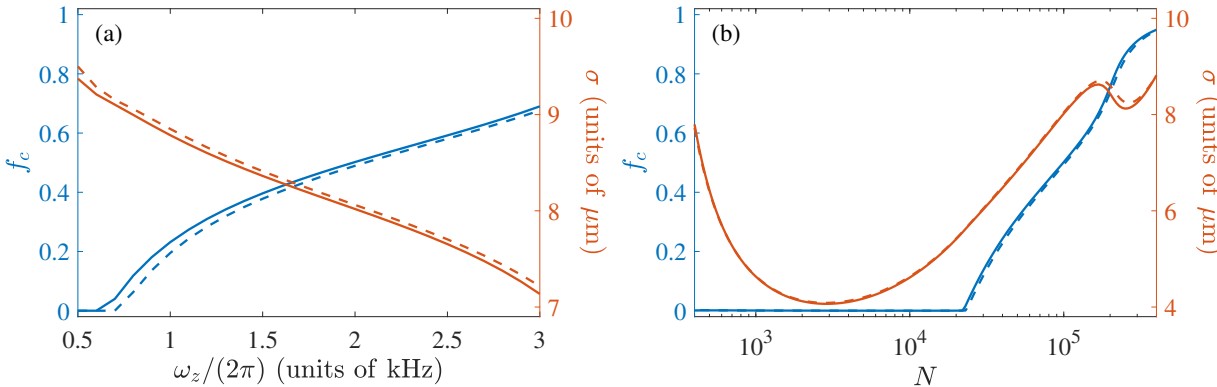

FIG. 1. Comparison between the full 3D (solid lines) and the quasi-2D calculations (dashed lines). (a) $\omega_z$ dependence of $f_c$ (blue lines) and $\sigma$ (red lines) for $N = 10^5$ and $\varepsilon_{\rm dd} = 0.6$. (b) $N$ dependence of $f_c$ (blue lines) and $\sigma$ (red lines) for $\omega_z = (2\pi)2\,{\rm kHz}$ and $\varepsilon_{\rm dd} = 0.6$. Other parameters are the same as those used in Sec. III C.

From Eqs. (16) to (17), it is clear that, in GST, the quantum fluctuation is incorporated into the wave functions through squeezed component. In fact, Gaussian states are vacuum states of Bogoliubov excitations. Furthermore, the quantum nature of the condensate can now be characterized as follows. A conventional condensate described by a CS should satisfy $N_c/N \approx 1$ and $N_s/N \ll 1$, where the squeezed atoms are the quantum depletion. In the opposite limit with $N_{s,1}/N \approx 1$, $N_c/N \approx 0$, and $N_{s,j>1}/N \approx 0$, the condensate is in a macroscopic single-mode squeezed vacuum state, i.e., SVS, which, as shown previously [33, 37], possesses completely different statistical properties compared to CS. Finally, for the intermediate case, both $\bar{\phi}_c$ and $\bar{\phi}_{s,1}$ are macroscopically occupied, the condensate is in SCS.

Before presenting our results, let us briefly compare GST with HFBT. At first sight, GST may seem equivalent to HFBT since, for a steady state, they yield the same mean value of field operator $\phi$ and the same correlation functions $G$ and $F$. However, it is well-known that HFBT suffers a issue regarding the gapped excitation spectrum which violates the Hugenholtz-Pines theorem [42–44]. The underlying reason is that HFBT only considers the variation of $\phi$. On the other hand, GST also takes into account the variations of $G$ and $F$, in addition to that of $\phi$. As a result, GST gives rise to the gapless excitation spectrum. [36–38].

## III. RESULTS

In this section, we shall first propose a scheme to create the quasi-2D droplets by tuning the sign of the DDI via a rotating polarization field. We then validate our quasi-2D model by comparing the results with full 3D calculation [33]. Finally, we study the ground-state properties of the droplet, including the phase diagram, the peak density, and the radial size. All numerical results are obtained by evolving Eqs. (14) with the fourth-order Runge-Kutta method. To Simplify our calculations we

also make use of the cylindrical symmetry of the system as in Ref. [33].

### A. Tuning DDI

Naively, one may think that quasi-2D dipolar droplets can be created by imposing a strong confinement along the $z$ direction. However, because DDI always stretches the condensate along the direction of the polarized dipoles, i.e., the $z$ axis, to minimize DDI energy, experimentally realized dipolar droplets are always of filament shape [45]. Fortunately, the sign of DDI can be tuned through a fast rotating polarization field around the $z$ axis such that the magnetic DDI between atoms becomes [46]

$$\widetilde{U}_{\rm dd}(\mathbf{k}_\rho) \to \chi \widetilde{U}_{\rm dd}(\mathbf{k}_\rho) \tag{19}$$

where $\chi$ is a factor continuously tunable between $-1/2$ and 1. For a negative $\chi$, DDI is attractive in the $x$-$y$ plane and repulsive along the $z$ axis. Under this type of anisotropy, it is energetically favorable to form the pancake-shaped droplets.

To analyze the stability of our system, we consider a quasi-2D homogeneous gas in the absence of the 3B interaction. For a negative $\chi$, the excitation spectrum takes the form [39]

$$\varepsilon_{\mathbf{k}_\rho} = \sqrt{\varepsilon_{\mathbf{k}_\rho}^0 \left\{ \varepsilon_{\mathbf{k}_\rho}^0 + 2n_0 \frac{4\pi\hbar^2 a_s}{\sqrt{2\pi}a_z M} \left[ 1 - \varepsilon_{\rm dd} D\left( \frac{|\mathbf{k}_\rho| a_z}{\sqrt{2}} \right) \right] \right\}}, \tag{20}$$

where $\varepsilon_{\mathbf{k}_\rho}^0 = \hbar^2 \mathbf{k}_\rho^2/(2M)$ is the spectrum of a free particle, $n_0$ is the density of the gas, and $\varepsilon_{\rm dd} \equiv |\chi| a_{\rm dd}/a_s$ is the relative dipolar interaction strength. By noting that $D(0) = 2$, Bogoliubov excitation becomes unstable in the small momentum limit when $\varepsilon_{\rm dd} > 1/2$. Therefore, in order to study the self-bound quasi-2d dipolar droplets, we shall always focus on the unstable regime with $\chi < 0$ and $\varepsilon_{\rm dd} > 1/2$.

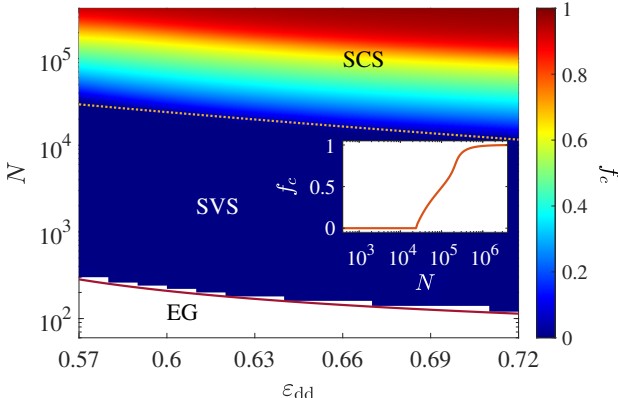

FIG. 2. Phase diagram on the $\varepsilon_{\mathrm{dd}}$-$N$ plane. The dotted line is the boundary between the SVS and the SCS phases and solid line marks the CAN $N_{\mathrm{cri}}^{(s)}$ analytically computed using Eq. (23). The inset shows the $N$ dependence of $f_c$ for $\varepsilon_{\mathrm{dd}} = 0.6$.

## B. Validity of the quasi-2D model

Since solving Eqs. (14) for a 3D system requires tremendous computing resource, it is crucial for us to utilize the quasi-2D geometry for a systematic study of the ground-state properties. To this end, we numerically solve our system via the full 3D and the quasi-2D calculations. We then compare the physical quantities computed from both approaches. In particular, we focus on the coherent fraction $f_c = N_c/N$ and the radial size $\sigma = \left[ \int d\boldsymbol{\rho} \rho^2 n(\boldsymbol{\rho})/N \right]^{1/2}$, where $n(\boldsymbol{\rho}) = \langle \hat{\psi}^\dagger(\boldsymbol{\rho})\hat{\psi}(\boldsymbol{\rho}) \rangle = |\phi(\boldsymbol{\rho})|^2 + G(\boldsymbol{\rho}, \boldsymbol{\rho})$ is the total density of the condensate. In Fig. 1(a), we plot the $\omega_z$ dependence of $f_c$ and $\sigma$. As can be seen, although there exists obvious discrepancy between two approaches at small $\omega_z$, it becomes negligibly small for sufficiently large $\omega_z$. Moreover, under the given trap frequency $\omega_z = (2\pi)2\,\mathrm{kHz}$, we compare the results from both approaches by varying $N$. As plotted in Fig. 1(b), the results from both approaches exhibit excellent agreement. Of particular importance, this agreement is insensitive to the variation of $N$. These results clearly show that our system is accurately described by the quasi-2D model when the one-dimensional confinement is sufficiently strong. Therefore we can safely use the quasi-2D equations for the remaining calculations.

## C. Ground-state properties

Here we study the ground-state properties of a self-bound quasi-2D droplet by varying the control parameters $N$ and $a_s$ (or equivalently, $\varepsilon_{\mathrm{dd}}$). To this end, let us first fix the values of other parameters to simplify our calculations. For the dipolar interaction length $|\chi|a_{\mathrm{dd}}$, without loss of generality, we choose $\chi = -0.155$ such that $|\chi|a_{\mathrm{dd}} = 20\,a_B$. Then, for the 3B coupling strength, we use $g_3 = 6.73 \times 10^{40}\hbar\mathrm{m}^6/\mathrm{s}$, the value fitted in our

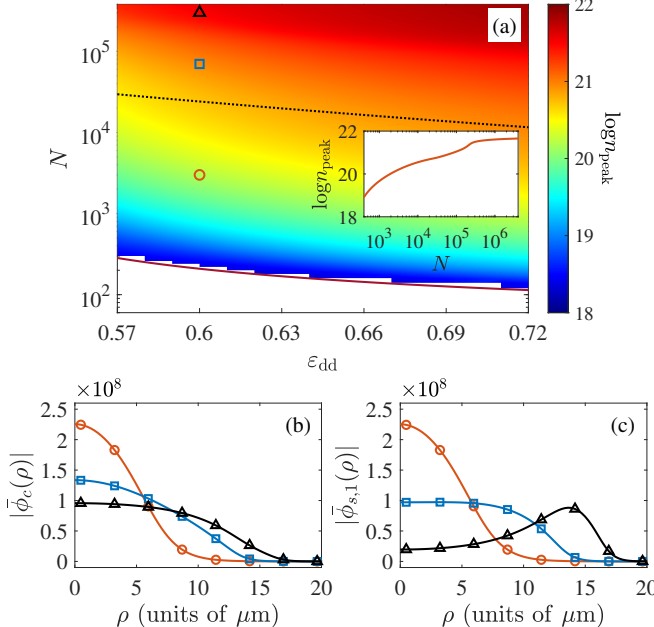

FIG. 3. (a) Distribution of the peak condensate $n_{\mathrm{peak}}$ (units of m$^{-3}$) on the $\varepsilon_{\mathrm{dd}}$-$N$ plane. The dotted line is the boundary between the SVS and the SCS phases and the solid line is the CAN calculated using Eq. (23). Markers denote the parameters used to plot the mode functions in (b) and (c). The inset shows the $N$ dependence of $\log n_{\mathrm{peak}}$ for $\varepsilon_{\mathrm{dd}} = 0.6$. (b) the coherent mode $|\bar{\phi}_c(\rho)|$ and (c) the first squeezed mode $\bar{\phi}_{s,1}(\rho)$ for $N = 3 \times 10^3$ ($\bigcirc$), $7 \times 10^4$ ($\square$), and $4 \times 10^5$ ($\triangle$) with $\varepsilon_{\mathrm{dd}} = 0.6$.

earlier work [33]. Finally, we fix the axial frequency at $\omega_z = (2\pi)2\,\mathrm{kHz}$, a value that can be readily achieved in experiments.

In Fig. 2, we map out the coherent fraction $f_c$ in the $\varepsilon_{\mathrm{dd}}$-$N$ parameter plane. The phase diagram is divided into three regions, resembling those of the 3D dipolar and the quasi-2D binary droplets except for that the CS phase is now absent. More specifically, the region below the CAN (solid line) represents the expanding gas (EG) where the attractive interaction is weak and insufficient for the formation of self-bound states. The middle and upper regions represent the self-bound SVS and SCS phases, respectively. In both phases, only one squeezed mode is macroscopically occupied, therefore they belong to the macroscopic squeezed states. However, compared to the SVS phase, the coherent fraction in the SCS phase is nonzero which breaks the $Z_2$ symmetry possessed by the SVS phase [37]. Similar to the 3D case, the transition between the SVS and SCS phases is also of third order [33].

Next, we inspect the density profile of the droplets. Fig. 3(a) shows the total peak density, $n_{\mathrm{peak}} = n(0)/(\sqrt{\pi}a_z)$, in the $\varepsilon_{\mathrm{dd}}$-$N$ parameter plane, where we have utilized that $n(\boldsymbol{\rho})$ always reaches maximum at the center (i.e., $\boldsymbol{\rho} = 0$) of the droplet, an observation found in our numerical results. Similar to the distribution of $f_c$,

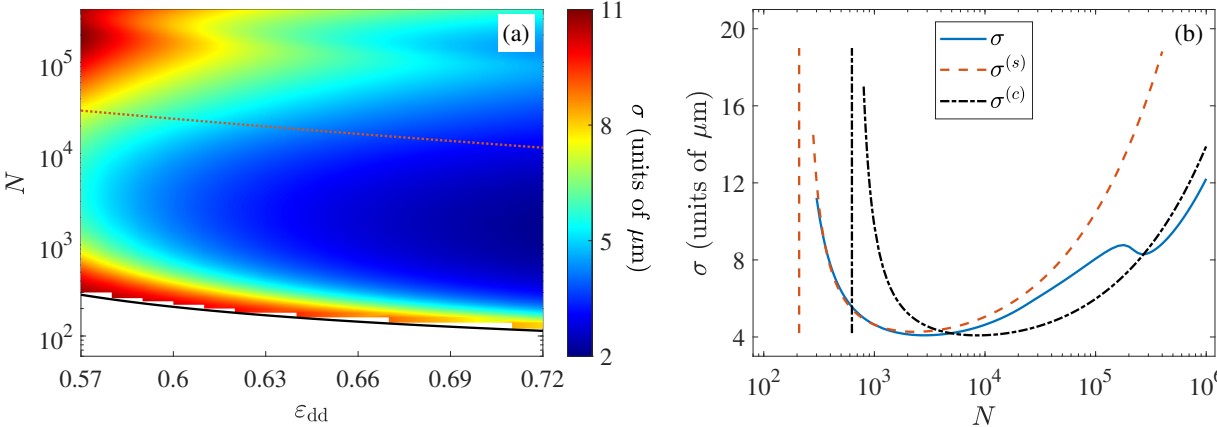

FIG. 4. (a) Distribution of the radial size $\sigma$ (units of $\mu$m) on the $\varepsilon_{\mathrm{dd}}$-$N$ plane. The dotted line is the boundary between the SVS and the SCS phases and the solid line is the CAN calculated using Eq. (23). (b) Radial size versus mean atom number for $\varepsilon_{\mathrm{dd}} = 0.6$. From left to right, two vertical lines mark the positions of $N_{\mathrm{cri}}^{(s)}$ and $N_{\mathrm{cri}}^{(c)}$, respectively.

$n_{\mathrm{peak}}$ is simply a monotonically increasing function of both variables, $\varepsilon_{\mathrm{dd}}$ and $N$. In particular, as shown in the inset of Fig. 3(a), $n_{\mathrm{peak}}$ saturates for a sufficiently large $N$. To gain more insight, we plot the coherent mode and the first squeezed mode in Fig. 3(b) and (c), respectively. In the SVS phase, $\bar{\phi}_c$ and $\bar{\phi}_{s,1}$ are visually identical to each other. This can be mathematically understood by noting that, in both SVS and SCS phases, $\bar{\phi}_c$ and $\bar{\phi}_{s,1}$ are described by a set of coupled two-mode equations [33]. It can then be verified that $\bar{\phi}_c = \bar{\phi}_{s,1}$ is a solution of those equations in the $N_c/N \approx 0$ limit. Then, following the increase of atom number (or total density), the 3B repulsion in the high density regime starts to turn the squeezed atoms into coherent ones. As a result, $\bar{\phi}_{s,1}$ is flattened. In addition, $\bar{\phi}_c$ is also flattened due to the repulsive 3B interaction. Finally, as the density is further increased, the $\bar{\phi}_{s,1}$ is significantly suppressed around the center of the droplet where the condensate has the highest density and the strongest 3B repulsion. Consequently, dip emerges on $\bar{\phi}_{s,1}$ around the center of the droplet.

We now study the properties of the radial size of the droplets. In Fig. 4(a), we present the distribution of the radial size $\sigma$ on the $\varepsilon_{\mathrm{dd}}$-$N$ plane, which, unlike Figs. 2 and 3(a), clearly exhibits features also shown in the binary droplets [34]. To reveal more details, we plot, in Fig. 4(b), the atom number dependence of the radial size under a fixed $\varepsilon_{\mathrm{dd}}$. As shown by the solid line, the $\sigma$-$N$ curve is of W shape, resembling that in the quasi-2D binary droplets.

Following Ref. [34], we analyze the $\sigma$-$N$ curve using a simple variational method. Given the radial size $\sigma$ of the droplet, the total energy per atom, $\epsilon = E/N$, can be expressed as

$$\epsilon(\sigma) \propto \frac{1}{\sigma^2} + \frac{[\tilde{g}_s + \tilde{g}_d f(\sigma)]N}{\sigma^2} + \frac{\tilde{g}_3 N^2}{\sigma^4}. \qquad (21)$$

where the terms on the right hand side are contributed by the kinetic energy, the 2B interaction energy, and

the 3B interaction energy, respectively. Moreover, $\tilde{g}_s$, $\tilde{g}_d$, and $\tilde{g}_3$ represent the reduced interaction strengths (RIS) for the contact, dipolar, and 3B interactions, respectively. $f(\sigma)$ accounts for the remaining $\sigma$ dependence in dipolar interaction energy. To find the explicit expression of $f(\sigma)$, we assume that the density profiles are described by Gaussian functions. To further simplify the calculation, we focus on two limiting cases: i) a pure coherent state with $\phi(\boldsymbol{\rho}) = \sqrt{\frac{N}{\pi\sigma^2}}e^{-\rho^2/(2\sigma^2)}$ and $G = F = 0$; ii) a single-mode squeezed vacuum state with $\phi = 0$ and $F(\boldsymbol{\rho}, \boldsymbol{\rho}') \approx G(\boldsymbol{\rho}, \boldsymbol{\rho}') = \frac{N}{\pi\sigma^2}e^{-(\rho^2+\rho'^2)/(2\sigma^2)}$. RIS can then be evaluated to yield $\tilde{g}_s^{(c)} = 2a_s/(\sqrt{2\pi}a_z)$, $\tilde{g}_d^{(c)} = 2a_{\mathrm{dd}}/(\sqrt{2\pi}a_z)$, and $\tilde{g}_3^{(c)} = g_3/(9\sqrt{3}\pi^3 a_z^4)$ for case i) and $\tilde{g}_s^{(s)} = 3\tilde{g}_s^{(c)}$, $\tilde{g}_d^{(s)} = 3\tilde{g}_d^{(c)}$, and $\tilde{g}_3^{(s)} = 15\tilde{g}_3^{(c)}$ for case ii). Moreover, the function $f$ takes the familiar form [47]

$$f(\sigma) = \frac{1}{\kappa^2 - 1}\left[2\kappa^2 + 1 - \frac{3\kappa^2 \tan^{-1}\sqrt{\kappa^2 - 1}}{\sqrt{\kappa^2 - 1}}\right],$$

where $\kappa = \sigma/a_z$.

Now, for a given set of RIS, the equilibrium radial size can be determined by numerically minimizing $\epsilon$. In Fig. 4(b), we plot the $N$ dependence of the equilibrium radial size $\sigma^{(c)}$ and $\sigma^{(s)}$ corresponding to cases i) and ii), respectively. As can be seen, both $\sigma^{(c)}(N)$ and $\sigma^{(s)}(N)$ are of V shape. In addition, $\sigma^{(s)}$ is in good agreement with the numerically computed $\sigma$ in the SVS phase regime and $\sigma^{(c)}$ also roughly agrees with $\sigma$ in the large $N$ limit. Since different quantum phases correspond to distinct $\sigma(N)$, the W-shaped $\sigma$-$N$ curve is clearly a manifestation that there are multiple quantum phases in the self-bound droplet state.

Finally, making use of the fact that $\lim_{\sigma\to\infty} f(\sigma) = -2$, we can derive an analytical expression for the CAN. To this end, we first note that both the kinetic and the 2B interaction energies in Eq. (21) scale as $\sigma^{-2}$ in the $\sigma \to \infty$ limit. Therefore, in the absence of the 3B interaction, the

kinetic and the 2B interaction energies exactly cancel out each other when $N$ satisfies

$$\frac{1}{\sigma^2} + \frac{(\tilde{g}_s - 2\tilde{g}_d)N}{\sigma^2} = 0. \tag{22}$$

This implies that, in the large $\sigma$ limit, $\epsilon(\sigma)$ is invariant with respect to $\sigma$. As a result, we find a self-bound droplet solution of infinity radial size which has the minimal atom number (i.e., CAN),

$$N_{\text{cri}} = \frac{1}{2\tilde{g}_d - \tilde{g}_s}. \tag{23}$$

It follows from the above analysis that the 3B repulsion would not change CAN, i.e., Eq. (23), since it decays much faster as $\sigma$ increases. In other words, because the radial size of the self-bound droplets in 2D geometry can be of infinity, CAN is independent of the 3B repulsion or any other stabilization forces. Interestingly, $N_{\text{cri}}$ still depends on the quantum state of the droplets via RIPs. In fact, the CANs $N_{\text{cri}}^{(c)}$ and $N_{\text{cri}}^{(s)}$ for the pure coherent state and the pure squeezed state satisfy a simple relation $N_{\text{cri}}^{(c)} = 3N_{\text{cri}}^{(s)}$. In Fig. 4(b), the vertical lines mark the positions of $N_{\text{cri}}^{(s)}$ and $N_{\text{cri}}^{(c)}$, which are in a good agreement with the numerical results. We point out that the independency of $N_{\text{cri}}$ on the stabilization mechanism makes it an ideal experimental criterion for quantum phases in droplets.

## IV. CONCLUSION

In conclusion, we have studied the ground-state properties of a quasi-2D self-bound dipolar droplet. We show that there exist two macroscopic squeezed phases. As a result, the radial size versus the atom number curve exhibits double-dip structure, which can be used as a qualitative signature for the multiple quantum phases in the self-bound droplet state. In addition, we find that the CAN for the self-bound droplets is determined by the quantum states and is independent of the stabilization mechanism. Therefore, the precise measurement of the CAN enables us to distinguish the quantum states and validates the GST. Compared to the Bose-Bose mixtures which may involve four unknown 3B coupling constants, dipolar condensates represent a much simpler configuration. Thus the proposed system allows us to have an in-depth study on ground-state phases of the quasi-2D droplets.

## ACKNOWLEDGMENTS

This work was supported by NSFC (Grants No. 12135018, No. 11974363, and No. 12047503), by the National Key Research and Development Program of China (Grant No. 2021YFA0718304), and by the Strategic Priority Research Program of CAS (Grant No. XDB28000000).

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
