# Peer review of "Self-bound droplets in quasi-two-dimensional dipolar condensates"

_SciPost Physics_

## Round 1 · Referee Report · Anonymous · 2022-12-23

Report

The authors present a study of the ground-state properties of self-bound dipolar droplets using Gaussian state theory (GSI), a theory which one of the authors also helped to develop. GSI is proposed as a useful theoretical tool to go beyond the usual modeling of such droplets based on the extended Gross-Pitaevskii equation (eGPE) and Lee-Huang-Yang corrections (LHY).

The main result of the paper is the identification and analysis of two quantum phases, corresponding to macroscopic squeezed vacuum and squeezed coherent states.

However, based on the fact that the eGPE is clearly limited I was also hoping for concrete examples where GSI can indeed "do better". Such evidence is not provided in this paper and hence the relevance of this new theoretical approach remains unclear to me. This is unfortunate because there are a number of experimental measurements (as cited by the authors) that could be used to directly compare and validate new theoretical predictions.

In this context I also want to note that the geometry assumed (I presume to make the approach tractable?) is far from any experimentally relevant situation. I am particularly wondering if the authors have checked that the extended 2D geometry studied indeed leads to individual self-bound droplets or rather crystalline states with many droplets in the eGPE? Knowing that the eGPE qualitatively describes dipolar BECs very well, it could easily be used to cross-check if the proposed droplet filaments appear in the parameter regime considered. Even if "true" comparisons with experiments are not yet feasible, this could help to put the new theory into better context.

In general, I find the narrative and conclusions of the manuscript challenging to follow. For example, what is the motivation to introduce three-body interactions in the modeling? It is my understanding that these have been experimentally ruled out beyond any doubt (see Ref. 21) in favor of LHY corrections? I believe any new theoretical model should thus - broadly speaking - refine LHY, rather than invoke a completely different mechanism. I did not find any supporting evidence as to why is is done here?

Most of the relevant literature is referenced. Note, though, that the tunability of dipolar interactions with a rotating field has already been demonstrated experimentally (Tang et al., PRL 116, 215301, 2018). This work should be cited.

In summary, the results presented in the manuscript are not entirely convincing and further examples and explanations are needed to fully understand the implications of the findings.

  • validity: -
  • significance: -
  • originality: -
  • clarity: -
  • formatting: -
  • grammar: good

Author:  Yuqi Wang  on 2023-04-06  [id 3555]

(in reply to Report 1 on 2022-12-23)
Category:
answer to question
reply to objection

However, based on the fact that the eGPE is clearly limited I was also hoping for concrete examples where GST can indeed "do better". Such evidence is not provided in this paper and hence the relevance of this new theoretical approach remains unclear to me. This is unfortunate because there are a number of experimental measurements (as cited by the authors) that could be used to directly compare and validate new theoretical predictions.

In the present work, we consider the pancake-shaped droplets formed by negating the usual dipole-dipole interaction (DDI) via a rotating orientation field. Since this is a new configuration for dipolar droplets, there is not any experimental realization. However, in the earlier studies of dipolar and binary droplets using GST [1,2], we have extensively compared our numerical results with the experimental measurements [3-5]. In below, we summarize these comparisons for both dipolar and binary droplets.

  • Particle number distributions (PNDs). In Refs.[3,4], the particle number distributions are measured. Surprisingly, all PNDs are asymmetric with respect to the mean atom number, which is in contradiction with the coherent-state assumption of the droplets. Although it is argued that this asymmetry may be caused by the finite temperature, the extremely longer tail in the PND reported in the Supplemental Material of Ref.[4] suggests a large atom number fluctuation unexplainable by the finite-temperature effect. On the other hand, these asymmetric PNDs appear naturally in GST. And, in Ref.[1], we fitted the PND obtained with GST and found good agreement with experimental data [3].
  • Radial size. In Ref.[2], we calculated the radial size of quasi-two-dimensional binary droplets and compared it with the experiment measurements [5]. In particular, we predicted that the radial-size-versus-atom-number curves are of the "W" shape due to the multiple quantum phases. These predictions are in striking contrast to the "V" shape predicted by EGPE and can in principle be tested in experiments.
  • Critical atom number. In study Ref. [2], we compared the calculated critical atom number with the experimental values [5], which exhibit good agreement. In addition, we find that the critical atom number of the quasi-two-dimensional droplets does not provide too much information for the stability mechanism of the droplets. Instead, the critical atom number is mainly determined by the quantum phase of the droplet.
  • Second-order correlation function. In Refs.[1,2], we also proposed to determine the quantum phases of the droplets by measuring the second-order correlation function, which can in principle be done in future experiments.

In this context I also want to note that the geometry assumed (I presume to make the approach tractable?) is far from any experimentally relevant situation. I am particularly wondering if the authors have checked that the extended 2D geometry studied indeed leads to individual self-bound droplets or rather crystalline states with many droplets in the eGPE? Knowing that the eGPE qualitatively describes dipolar BECs very well, it could easily be used to cross-check if the proposed droplet filaments appear in the parameter regime considered. Even if "true" comparisons with experiments are not yet feasible, this could help to put the new theory into better context.

The scheme we proposed in this work indeed leads to a single self-bound droplet. This result has also been confirmed by numerically solving EGPE.

Generally speaking, to lower the interaction energy, a dipolar condensate stretches (shrinks) along the attractive (repulsive) direction. As a result, a conventional dipolar droplet appears as a filament along the attractive ($z$) direction. In addition, because the transitional symmetry is prone to be broken by the repulsive interaction, crystalline state and isolated filaments only form on the repulsive ($xy$) plane.

In our configurations, the dipolar interaction is negated through a rotating orientation field such that it becomes attractive (repulsive) on the $xy$ plane (along the $z$ direction). Consequently, a dipolar condensate expands on the $xy$ plane and is of the shape of the pancake. Furthermore, the strong harmonic trap imposed on the repulsive $z$ direction prevents the formation of the crystalline state along the $z$ direction. As a result, the dipolar droplet studied in our work is a single self-bound one.

In general, I find the narrative and conclusions of the manuscript challenging to follow. For example, what is the motivation to introduce three-body interactions in the modeling? It is my understanding that these have been experimentally ruled out beyond any doubt (see Ref. 21) in favor of LHY corrections? I believe any new theoretical model should thus - broadly speaking - refine LHY, rather than invoke a completely different mechanism. I did not find any supporting evidence as to why is done here?

Let us first recall why the three-body (3B) repulsion was ruled out as a stabilization mechanism for quantum droplets. In 2015, Bisset and Blakie studied dipolar droplets by numerically solving GPE with 3B repulsion [6]. They estimated that the real and imaginary parts of the 3B coupling strength was $\kappa_r=5.87\times10^{-39}\hbar{\rm m^6/s}$. Then, in Ref.[7], it was experimentally determined that the imaginary part of the 3B coupling strength $\kappa_i$ was below $10^{-41}\hbar{\rm m^6/s}$, a value much smaller than $\kappa_r$. Since this is in contradiction with the speculation that $\kappa_r$ and $\kappa_i$ should be close to each other, the three-body interaction was then ruled out as the stabilization mechanism for dipolar droplets[7]. However, we re-estimated $\kappa_r$ using Gaussian-state theory and obtained $\kappa_r=6.73\times10^{40}\,\hbar{\rm m^6/s}$ [1], a value representing a big improvement compared to the GPE estimation. Consider that the recent experimentally measured $\kappa_i=1.33\times10^{-41}\,\hbar{\rm m^6/s}$ is larger than the earlier measurement [4], it is still too early to say that the 3B repulsion is ruled out.

To further justify the inclusion of the 3B repulsion, we note that, from a more fundamental level, the many-body Hamiltonian of the condensed matter physics are derived within an effective low energy theory, which, after integrating out the high energy excitations, gives rise to interaction terms involving two particles, three particles, and so on. Therefore, the three-body interaction intrinsically presents in our Hamiltonian. When we study a high density gas, like dipolar droplets, it is natural to include the 3B interaction. Moreover, given the large 3B loss rate observed in experiments [3,4], it is not surprising that the 3B repulsion should also play a role in the system.

Now, for the relation between EGPE and GST, let us briefly recall the derivation of EGPE. To start, we first solve the Gross-Pitaevskii equation (GPE) to find the wave function of the condensate $\phi({\mathbf r})$. Here we have implicitly assumed that the many-body quantum state of the condensate $|\Psi\rangle$ is a coherent state. Next, we introduce the quantum fluctuation $\delta\hat{\psi}({\mathbf r})$ and compute the Bogoliubov excitations for a homogeneous gas. When all Bogoliubov excitations are stable, we calculate the lowest order energy correction induced by the quantum fluctuation, i.e., the Lee-Huang-Yang (LHY) correction. EGPE can then be obtained by incorporating the LHY correction into the GPE through local-density approximation:

$$ i\hbar\frac{\partial\phi}{\partial t}=\left[-\frac{\hbar^2\nabla^2}{2m}+\int d{\mathbf r}'V({\mathbf r}-{\mathbf r}')|\phi({\mathbf r}')|^2+g_{\rm LHY}[\varepsilon({\mathbf k})]|\phi|^3\right]\phi, \tag{1}\label{egpe} $$

where the coefficient $g_{\rm LHY}[\varepsilon({\mathbf k})]$ is a functional of the Bogoliubov excitation $\varepsilon({\mathbf k})$. After quantum fluctuation is included, the many-body quantum state of the condensate $|\Psi\rangle$, i.e., the vacuum state of the Bogoliubov quasiparticles, is now a Gaussian state, in which quantum fluctuation is represented by the squeezed state. However, it should be noted that EGPE is still a coherent-state based theory since the quantum fluctuation is included preturbatively.

On the other hand, the Gaussian state ansatz is assumed for the many-body quantum state of the condensate in the GST [8]. Consequently, the first-order correlations, e.g., $\langle\delta\hat{\psi}^\dagger({\mathbf r}')\delta\hat{\psi}({\mathbf r})\rangle$ and $\langle\delta\hat{\psi}({\mathbf r}')\delta\hat{\psi}({\mathbf r})\rangle$, that are completely neglected in EGPE, are now included as order parameters besides $\phi({\mathbf r})$. These order parameters are self-consistently computed by minimizing the total energy, which gives rise to the many-body quantum state $|\Psi\rangle$. Of particular interest, as shown in our earlier work, EGPE can be derived from the equations of motion of the GST as the first-order perturbation (see the Supplemental Material of Ref.[9]). Compared to coherent state, Gaussian state covers a much larger variational parameter space. Consequently, the ground-state solution obtained in GST has lower energy under the same system configuration and is, therefore, more reliable. In Tab.1, we summarize the main difference between EGPE and GST.

Table 1: Comparison of EGPE and GST

<tbody> </tbody>
EGPE GST
Many-body-quantum state $\lvert\Psi\rangle$ is a coherent state|$\lvert\Psi\rangle$ is a Gaussian state, i.e., squeezed-coherent state $\lvert\Psi\rangle$ is a Gaussian state, i.e., squeezed-coherent state
Variational parameters $\phi({\mathbf r})\equiv\langle\Psi\rvert\hat{\psi}({\mathbf r})\lvert\Psi\rangle$ $\phi({\mathbf r})$, $\langle\Psi\rvert\delta\hat{\psi}^\dagger({\mathbf r}')\delta\hat{\psi}({\mathbf r})\lvert\Psi\rangle$, and $\langle\Psi\rvert\delta\hat{\psi}({\mathbf r}')\delta\hat{\psi}({\mathbf r})\lvert\Psi\rangle$ with $\delta\hat{\psi}({\mathbf r})=\hat{\psi}({\mathbf r})-\phi({\mathbf r})$
Quantum fluctuation perturbatively included self-consistently considered
Relation EGPE is a special case of GST and can be derived from the equations of motion of GST.
With that being said, let us analyze the validity of EGPE for the quantum droplet problems. Strictly speaking, Eq.\eqref{egpe} is not applicable in this regime since GPE is unstable in the parameter regime where quantum droplets emerge. Nevertheless, to describe the droplet state, Eq.\eqref{egpe} is analytically continued to the collapse regime where $g_{\rm LHY}$ becomes a complex parameter due to the imaginary Bogoliubov modes. For droplets in both dipolar and binary condensates, it can be verified that ${\rm Im}(g_{\rm LHY})$ is positive. Consequence, the result predicts by Eq.~\eqref{egpe} is obviously unphysical as atom number in the droplet would grow with time. To cure this problem, the imaginary part of $g_{\rm LHY}$ is simply neglected when Eq.\eqref{egpe} is used in the collapse regime.

Although it was argued that the EGPE should be applicable in the vicinity of the stability boundary. This claim, however, was never proven through self-consistent calculations with fluctuations being included. In addition, to simulate experiments, EGPE was often employed with parameters deep inside the collapse regime. For instance, in typical dipolar droplet experiments~\cite{[3]}, the reduced dipole interaction strength $\varepsilon_{\rm dd}$ can be as large as $2.5$, a value much larger than $\varepsilon_{\rm dd}=1$ that defines the stability boundary.

Still, one may insist that the correctness of EGPE was somehow demonstrated by the fact that it provides satisfactory explanation to experimentally measured critical atom numbers. However, as explained in Ref.[2] and also in the present work, the critical atom number is not a suitable quantity for identifying the stabilization mechanism for quantum droplets. In fact, in the experiment performed in Pfau's group, systematic discrepancy between the experimentally measured critical atom number and the EGPE simulations was observed [4]. In addition, it is important to note that some of the experimental results, such as the highly asymmetric atom number distributions [4], cannot be explained naturally by the EGPE theory.

Finally, we point out that since EGPE is a perturbative treatment of quantum fluctuation, its validity should be checked by the more comprehensive GST. In fact, our numerical simulations show that the EGPE results are in good agreement with those of GST in the mean-field stable regime where the coherent state is dominant. However, in the mean-field collapse regime, two approaches lead to completely different results.

Most of the relevant literature is referenced. Note, though, that the tunability of dipolar interactions with a rotating field has already been demonstrated experimentally (Tang et al., PRL 116, 215301, 2018). This work should be cited.

We thank the referee for reminding us about this reference. It is now being cited in the revised manuscript.

In summary, the results presented in the manuscript are not entirely convincing and further examples and explanations are needed to fully understand the implications of the findings.

We hope that the revised manuscript and our reply can now help the referee to understand the implications of our findings.

References

[1] Y. Wang, L. Guo, S. Yi, and T. Shi, Phys. Rev. Research $\bf2$, 043074 (2020).

[2] J. Pan, S. Yi and T. Shi, Phys. Rev. Research $\bf4$, 043018 (2022).

[3] M. Schmitt et al., Nature $\bf539$, 259 (2016).

[4] F. B$\ddot{\text{o}}$ttcher et al., Phys. Rev. Research $\bf1$, 033088 (2019).

[5] C. R. Cabrera et al., Science $\bf359$, 301 (2018).

[6] R. N. Bisset and P. B. Blakie, Phys. Rev. A $\bf92$, 061603 (2015).

[7] I. Ferrier-Barbut et al., Phys. Rev. Lett. $\bf116$, 215301 (2016).

[8] T. Shi, E. Demler, and J. I. Cirac, Ann. Phys. $\bf 390$, 245 (2018).

[9] T. Shi, J. Pan, and S. Yi, arXiv:1909.02432 (2019).

---

## Round 1 · Referee Report · Anonymous · 2023-1-12

Strengths

1- interesting and informative
2- clearly written

Weaknesses

1- physically wrong in spite of mathematical validity

Report

The authors use the Gaussian variational ansatz for analyzing self-bound states of quasi-two-dimensional bosonic dipoles. They find that in a region of parameters (in particular, for higher particle numbers N) the self-bound state is described by the usual BEC-type mean-field solution. In the Gaussian-ansatz language they call this limit squeezed coherent state. They claim that in another limit (smaller N) the coherent fraction vanishes and the droplet is described by a very different Gaussian state called single-mode squeezed vacuum. The authors previously discussed these two phases for bosons in other setups in Refs.[33-34,37].

After reading these papers I come to the conclusion that the single-mode squeezed vacuum state is an artificial mathematical construction which cannot be observed in practice. My arguments are as follows.

To my understanding we work with Gaussian states since they provide mathematical advantages (Wick theorem), at the same time not deviating too much from the number-conserving reality. The coherent state given by Eq.(16) with d_j=0 and N_c=<N> is suitable for describing the usual BEC since for large N_c it is very close to the Fock state with N=N_c, fluctuations of the atom number being ~Sqrt[N]<<N. Therefore, for both Fock and Gaussian states the moments are <N^2>=<N>^2 and <N^3>=<N>^3, which is important for quantifying the two-body and three-body interactions.

By contrast, the single-mode squeezed vacuum state asymptotically corresponds to Eq.(16) with N_c=0, Sinh^2(d_1)=<N>, and all other d_i=0. This state is a quantum superposition of Fock states with very different N. The number distribution is peaked at 0, not at N=<N>. Apparently, the moments of this distribution equal <N^2>=3<N>^2 and <N^3>=15<N>^3 (for comparison, an equal-weight superposition of Fock states with N=0 and N=2<N> corresponds to <N^2>=2<N>^2 and <N^3>=9/2<N>^3). By the way, this is why the Gross-Pitaevskii energy-density functional for the spatial mode function in the squeezed-vacuum limit looks like the one written for the usual BEC, but with the two-body and three-body coupling constants replaced by g_2->3g_3 and g_3->15g_3.

So, for a single-mode squeezed vacuum state corresponding to a certain fixed <N>, the interaction energy is dominated by the contribution of Fock states with significantly higher atom numbers. In this respect one deals with an object with effectively higher N. Given that self-bound droplets gain energy when they absorb particles, it is thus not surprising that the energy minimization procedure may prefer the single-mode squeezed construction to the coherent state formally corresponding to the same <N>. I should mention that Eq.(21), even neglecting the dipolar term g_d, is very useful for verifying that the single-mode squeezed vacuum state can indeed become lower in energy than the coherent state... provided that fixing <N> is physically meaningful.

Unfortunately, this last point is problematic. Why do the authors think that in practice the energy should be minimized at fixed <N>? Imagine that N atoms are prepared in the experiment. The system then has to eject some of them in order to get into the quantum superposition of different Fock states required by the single-mode squeezed vacuum state. I suspect that it will rather prefer to keep all N atoms and get into the usual N-body BEC state.

One can also assume that the system is somehow connected to a bath (of thermal atoms?) However, the problem remains the same. Since atom numbers significantly larger than the desired <N> are available, from my viewpoint, the system would just exhaust the bath and fall into a Fock state with maximum available N. In any case, the minimization should not be done at fixed <N>, but at fixed total N (droplet plus bath), at fixed chemical potential, or something of this kind depending on the concrete setup.

Even if I accept the authors' mathematical formulation of the problem which implies minimization at fixed <N> (which I find artificial), is there a reason why one should look for the solution only among Gaussian states? Most likely there is another superposition of Fock states, which profits from the two-body and three-body interactions in a more optimal manner than the single-mode vacuum state.

Although I enjoyed the paper, I cannot recommend it for publication. I hope that my concern is clear. How do the authors ensure the constraint <N>=const, at the same time allowing for large number fluctuations in a self-bound object? I strongly suspect that in any real experiment the usual BEC state will always win.

  • validity: -
  • significance: top
  • originality: ok
  • clarity: top
  • formatting: perfect
  • grammar: -

Author:  Yuqi Wang  on 2023-04-06  [id 3554]

(in reply to Report 2 on 2023-01-12)

Strengths: 1 interesting and informative 2 clearly written Weaknesses: 1 physically wrong in spite of mathematical validity

We thank the referee for the brief summary for our work. Although the referee agrees that our work is mathematically correct, he/she is concerned about the physical validity of the single-mode squeezed vacuum (SMSV) state. Before reply to the comments/questions of the referee's, let us first summarize the main properties of the SMSV state.

The SMSV state only emerges when the overall two-body interaction is attractive and the three-body repulsion is negligible. Under these conditions, the two-body attraction is balanced by the kinetic energy such that a condensate of the SMSV state can only sustain a small number of atoms (say, several thousands, depending on the $s$-wave scattering length). Otherwise, it would either collapse (in the absence of the three-body repulsion) or undergo a phase transition to the squeezed coherent state (in the presence of the three-body repulsion).

After reading these papers I come to the conclusion that the single-mode squeezed vacuum state is an artificial mathematical construction which cannot be observed in practice. My arguments are as follows.

To my understanding we work with Gaussian states since they provide mathematical advantages (Wick theorem), at the same time not deviating too much from the number-conserving reality. The coherent state given by Eq.(16) with $d_j=0$ and $N_c=\langle N\rangle$ is suitable for describing the usual BEC since for large $N_c$ it is very close to the Fock state with $N=N_c$, fluctuations of the atom number being $\sqrt{N}\ll N$. Therefore, for both Fock and Gaussian states the moments are $\langle N^2\rangle=\langle N\rangle^2$ and $\langle N^3\rangle=\langle N\rangle^3$, which is important for quantifying the two-body and three-body interactions.

Since the referee accepts the coherent state description of the Bose-Einstein condensates, he/she agrees that, to find the ground-state wave function of a condensate, one should minimize the total energy at the fixed mean atom number $\langle N\rangle$ since the coherent state coherent state assumption breaks the $U(1)$ gauge symmetry of the system. The only concern (as indicated in the referee report below) of the referee is that the fluctuation for a SMSV state is significantly larger than that of a coherent state with the same $\langle N\rangle$.

By contrast, the single-mode squeezed vacuum state asymptotically corresponds to Eq.(16) with $N_c=0$, $\sinh^2(d_1)=\langle N\rangle$, and all other $d_i=0$. This state is a quantum superposition of Fock states with very different N. The number distribution is peaked at 0, not at $N=\langle N\rangle$. Apparently, the moments of this distribution equal $\langle N^2\rangle=3\langle N\rangle^2$ and $\langle N^3\rangle=15\langle N\rangle^3$ (for comparison, an equal-weight superposition of Fock states with $N=0$ and $N=2\langle N\rangle$ corresponds to $\langle N^2\rangle=2\langle N\rangle^2$ and $\langle N^3\rangle=9/2\langle N\rangle^3$).

The referee is correct on these results.

By the way, this is why the Gross-Pitaevskii energy-density functional for the spatial mode function in the squeezed-vacuum limit looks like the one written for the usual BEC, but with the two-body and three-body coupling constants replaced by $g_2\to3g_3$ and $g_3\to15g_3$.

The referee is correct on this. In fact, independent of the many-body wave function (e.g., coherent state, squeezed vacuum state, or other superpositions of the Fock states), the Gross-Pitaevskii energy-density functional for the spatial mode function always has the same form as the usual BEC as long as only a single spatial mode is involved.

So, for a single-mode squeezed vacuum state corresponding to a certain fixed $\langle N\rangle$, the interaction energy is dominated by the contribution of Fock states with significantly higher atom numbers. In this respect one deals with an object with effectively higher $N$.

In addition to the atom numbers in the Fock states, the contribution to the interaction energy also depends on the amplitudes of the Fock states in the superposition. More precisely, the interaction energy can be expressed as the function of $\langle N\rangle$. For the last sentence, if we understand correctly, what the referee really means is "... an object with effectively higher $\langle N\rangle$".

Given that self-bound droplets gain energy when they absorb particles, it is thus not surprising that the energy minimization procedure may prefer the single-mode squeezed construction to the coherent state formally corresponding to the same $\langle N\rangle$.

The referee is correct on that the SMSV state has a lower energy due to the factor-of-three enhancement on the interaction energy.

I should mention that Eq.(21), even neglecting the dipolar term $g_d$, is very useful for verifying that the single-mode squeezed vacuum state can indeed become lower in energy than the coherent state... provided that fixing $\langle N\rangle$ is physically meaningful.

Since the requirement for realizing the SMSV state is that the overall interaction is attractive, it is correct that we may neglect the dipolar interaction provided that the $s$-wave scattering is negative.

Unfortunately, this last point is problematic. Why do the authors think that in practice the energy should be minimized at fixed $\langle N\rangle$? Imagine that N atoms are prepared in the experiment. The system then has to eject some of them in order to get into the quantum superposition of different Fock states required by the single-mode squeezed vacuum state. I suspect that it will rather prefer to keep all N atoms and get into the usual N-body BEC state.

Strictly speaking, instead of being a Fock state, the usual $N$-body BEC state is a coherent state as it is widely accepted that the $U(1)$ gauge symmetry is broken when the system undergoes the Bose-Einstein condensation.

Given that $N$ atoms are initially prepared in an experiment, the superposition of the Fock states can be physically realized via the evaporation cooling and/or three-body inelastic collisions. In this sense, the SMSV state which also breaks the $U(1)$ gauge symmetry does not differ too much from the coherent state except for that the former has a much larger atom number fluctuation. However, because the SMSV state only exists for condensates with a small atom number, this atom number fluctuation should not cause too much trouble.

One can also assume that the system is somehow connected to a bath (of thermal atoms?) However, the problem remains the same. Since atom numbers significantly larger than the desired $\langle N\rangle$ are available, from my viewpoint, the system would just exhaust the bath and fall into a Fock state with maximum available $N$.

The reason that the bath would be exhausted is that the overall atom-atom interaction is attractive. As a result, it is energetically favorable to have more atoms in the system. In fact, the same thing would happen even for an usual BEC described by the coherent state.

However, for a real system with a sufficiently large bath, the wave function of the system cannot be a Fock state with maximum available atom number, since the condensate would either collapse in the absence of the three-body repulsion or stop absorbing atoms from the bath when it costs energy due to the three-body repulsion increases.

In any case, the minimization should not be done at fixed $\langle N\rangle$, but at fixed total N (droplet plus bath), at fixed chemical potential, or something of this kind depending on the concrete setup.

The referee suggests to minimize the energy 1) at fixed total $N$; 2) at fixed chemical potential; 3) by following the concrete experimental setup. In below, we give a brief discussion on each of them.

For suggestion 1), we are not aware of any tractable theoretical method that is capable of handling a system of at least several thousand atoms consisting of a thermal gas and a condensate.

For suggestion 2), we note that the chemical potential is introduced as a mathematical tool to fix the mean atom number in a grand cannonical ensemble. Therefore, there is no obvious physical reason that predetermines the value of the chemical potential for a bosonic system.

Finally, we believe suggestion 3) is somehow equivalent to minimizing energy at fixed $\langle N\rangle$. For a typical cold-atom experiment, a condensate are created after the laser cooling and the evaporation cooling. Experimentally, even if all control parameters in each step are fixed, the atom numbers in the condensates corresponding different runs of the experiments still fluctuate. As a result, the measured atom number satisfies certain statistical distribution which directly relates to the quantum nature of the state. For example, it was experimentally shown that the number distribution of a conventional BEC is a Possionian, in accordance with the coherent-state description of an usual condensate. The above analysis indicates that for a concrete experimental setup, one cannot fix the total atom number. Instead, the mean atom number in the condensate can be fixed.

Now, for the quantum droplet experiments, the three-body loss becomes prominent due to the large gas density, which gives rise to further uncertainty on atom number of the droplets. As a result, large atom number fluctuations were observed in Refs.[3,4]. Particularly, these atom number distributions are all asymmetric, in striking contrast to the symmetric ones of the usual condensates. As shown our earlier work using Gaussian state theory [1,2], the asymmetric atom number distributions can be naturally understood in the presence of the squeezed component in droplets.

Even if I accept the authors' mathematical formulation of the problem which implies minimization at fixed $\langle N\rangle$ (which I find artificial), is there a reason why one should look for the solution only among Gaussian states? Most likely there is another superposition of Fock states, which profits from the two-body and three-body interactions in a more optimal manner than the single-mode vacuum state.

In below, we list the main advantages of the Gaussian states. - The Gaussian states are generalization to the coherent states. Since any two-operator correlations are encoded into the variational parameters of a Gaussian state, the Gaussian state theory represents the most thorough mean-field theory. In this sense, Gaussian state theory is a natural candidate if one wants to consider a variational ansatz which includes higher order correlations beyond the conventional coherent-state-based theory for BEC. - The expectation values of physical observables can be efficiently computed via the Wick's theorem. This feature is highly nontrivial for an arbitrary many-body state. In addition, the Gaussian states also have a clear physical meaning after factorizing the wave function into the coherent and squeezed parts - Up to the first-order correlation functions level, the Gaussian state theory is equivalent to the Hartree-Fock-Bogoliubov theory (HFBT)~ [2]. However, unlike HFBT which gives rise to the gapped spectrum and violates the Hugenholtz-Pines theorem, the Gaussian state theory leads to the gapless excitation spectrum by taking into account the two-particle excitations.

One can certainly choose other ansatzes, however, we are not aware of any other wave functions that possess the same advantages as the Gaussian states.

Although I enjoyed the paper, I cannot recommend it for publication. I hope that my concern is clear. How do the authors ensure the constraint $\langle N\rangle$=const, at the same time allowing for large number fluctuations in a self-bound object? I strongly suspect that in any real experiment the usual BEC state will always win.

The last statement made by the referee seems rather. In fact, the experimental measurements (see, e.g., Fig. 6 in the Supplemental Material of Ref.[4]) clearly show that the self-bound droplets indeed exhibit a number fluctuation significantly larger than that of the usual BEC state. Although, in Ref.[4], this large fluctuation was attributed to the thermal gas which follows the Maxwell-Boltzmann distribution, it can be most naturally explained using the Gaussian state theory [1]. We point out that the origin of the large number fluctuation can be further identified in future experiments by comparing the number fluctuations of condensates prepared inside and outside the self-bound droplet regimes. Nonetheless, it is still too early to claim without further experiment evidences.

References

[1] Y. Wang, L. Guo, S. Yi, and T. Shi, Phys. Rev. Research $\bf2$, 043074 (2020).

[2] J. Pan, S. Yi and T. Shi, Phys. Rev. Research $\bf4$, 043018 (2022).

[3] M. Schmitt et al., Nature $\bf539$, 259 (2016).

[4] F. B$\ddot{\text{o}}$ttcher et al., Phys. Rev. Research $\bf1$, 033088 (2019).

---

## Round 2 · Author Response

Following this letter, we are resubmitting to you the revised manuscript entitled “Self-bound droplets in quasi-two-dimensional dipolar condensates” to be reconsidered for publication on SciPost Physics.
In the referee report, the first referee raised a few detailed questions and gave us some useful suggestions. We have responded to all questions of the referee and followed essentially all suggestions in the revised manuscript. Although the second referee accepted our mathematical formulation and enjoyed this work, he/she questioned the physical meaning of the work. In the reply, we response to his/her concern and revised the manuscript accordingly.
We strongly believe that our work is of SciPost Physics caliber and deserve a publication on SciPost Physics.
Yours sincerely,
Yuqi Wang, Tao Shi, Su Yi

---

## Round 2 · List of Changes

1. We add two sentences:
"Therefore, SVS is energetically favorable when the s-wave scattering length is negative. In the absence of the three-body repulsion, the two-body attraction is balanced by the kinetic energy such that a condensate of SVS can only sustain a small number of atoms before it undergoes a collapse [37]"
behind the third paragraph of the section I.
2. We add one sentence:
"We demonstrate that the macroscopic squeezed states, i.e., SVS and SCS, can be experimentally distinguished from other states by measuring the particle number distribution and the second-order correlation function."
in the fourth paragraph of the section I.
3. We add three sentences:
"It should be noted that the 3B interaction is not included in the Hamiltonian phenomenologically. From a more fundamental level, it originates from a low-energy effective theory after integrating out the high energy excitations. When we study a high-density gas, like quantum droplets, it is natural to include the 3B interaction."
behind the first paragraph of the subsection II.A.
4. We add four sentences:
"Compared to the coherent-state-based EGPE which perturbatively includes quantum fluctuation, GST takes into account quantum fluctuation self-consistently via the Gaussian state ansatz. As a result, it was shown that EGPE could be analytically derived from GST starting from a coherent state [37]. We also numerically verified that the EGPE results are in good agreement with those of GST if the ground state wave function is dominant by the coherent state [37]. Furthermore, let us briefly compare GST with HFBT."
before the last paragraph of the subsection II.B.
5. We add two sentences:
"It is worthwhile to mention that, for the negated DDI in quasi-2D geometry, we always obtain a single self-bound droplet in the GST calculations. In fact, this observation is also confirmed by the EGPE calculations."
behind the last paragraph of the subsection III.A.
6. Reference [47] Tang et al., PRL 120, 230401 has been added.
The changes are marked by red color.

---

## Editorial Decision

unknown